# Geographical distribution of antimicrobial exposure among very preterm and very low birth weight infants: A nationwide database study in Japan

**Kota Yoneda[1,2], Daisuke Shinjo[1]\*, Naoto Takahashi[2], Kiyohide Fushimi[1]**

**1** Department of Health Policy and Informatics, Tokyo Medical and Dental University Graduate School, Tokyo, Japan, **2** Department of Pediatrics, The University of Tokyo Hospital, Tokyo, Japan

\* dshinjo.hci@tmd.ac.jp

## Abstract

### Objectives

To examine spatial effects in neonatal care, we conducted a retrospective cohort study to investigate the geographical distribution of antimicrobial exposure among very preterm and very low birth weight infants in Japan.

### Study design

We utilized a nationwide claims database in Japan to extract prescriptions of injectable antimicrobials for 41,423 very preterm and very low birth weight infants admitted within the first two days of life from April 2010 to March 2021. We identified frequently prescribed antimicrobials, revealed early neonatal exposure and neonatal exposure to each antimicrobial agent by 47 prefectures in Japan, and evaluated their spatial autocorrelation using global and local Moran's *I* statistics. We then scrutinized regional disparities in antimicrobial drug prescriptions.

### Results

The top 10 antimicrobials prescribed to very preterm and very low birth weight infants in Japan were ampicillin, amikacin, gentamicin, cefotaxime, fluconazole, ampicillin combination, micafungin, cefmetazole, cefazolin, and vancomycin. We identified northern cold spots for fluconazole exposure and southern hot spots for ampicillin, amikacin, gentamicin, and cefmetazole exposure. Geographical heterogeneity in the selection of antibacterial and antimycotic agents was observed.

### Conclusion

Our study revealed the geographical distribution of antimicrobial exposure among very preterm and very low birth weight infants in Japan, thus disclosing its spatial effects. Further research addressing the spatial effects of neonatal care is needed to understand how drug exposure affects the outcomes of preterm infants.

**Data Availability Statement:** Data cannot be disclosed to the public due to a license agreement and ethical issues in each participating facility. To request the dataset generated during this study,

please contact the Office of Life Science and Bioethics Research Center via: Email: infobec@tmd.ac.jp Telephone: +81-3-3813-6111 The corresponding author is also available for data requests.

**Funding:** This study was supported by a Grant-in-Aid for Research on Policy Planning and Evaluation from the Ministry of Health, Labour and Welfare, Japan (grant number 22AA2003 [to KF]) and a Grant-in-Aid for Scientific Research (B) from the Japan Society for the Promotion of Science (JSPS KAKENHI, grant number 20H03921 [to DS]). The funders had no role in study design, data collection and analysis, decision to publish, or preparation of the manuscript.

**Competing interests:** The authors have declared that no competing interests exist.

**Abbreviations:** ATC, anatomical therapeutic chemical; DPC, diagnosis procedure combination; VPT, : very preterm; VLBW, very low birth weight.

## Introduction

Prematurity remains a major global burden, with preterm birth complications accounting for 0.94 million (17.7%) of 5.30 million under-five deaths worldwide in 2019 [1, 2]. Life-saving antibiotic administration for preterm infants can occasionally lead to adverse effects such as altered intestinal microbiota, acute kidney injury, and irreversible sensorineural hearing loss [3–6]. Clinicians should develop individualized treatment strategies based on drug safety information; however, limited data on the safety and efficacy of drugs in neonates are available, and 30 of the top 50 most frequently used medications in neonatal intensive care units in the United States were off-label [7–9]. While researchers must become current with the changing trends of medication use for preterm infants, the details of neonatal care in Japan have not been well-reported, despite the country having one of the lowest mortality rates for preterm infants [7, 10, 11].

Furthermore, antimicrobial administration in neonatal care may exhibit regional variations within the country because of the absence of well-established standards [12, 13]. Anselin defined spatial effects as spatial autocorrelation (or geospatial correlation within variables) and geospatial heterogeneity (or heteroscedasticity) [14–16]. Unmeasured confounding factors with spatial effects would violate the assumption of non-autocorrelation and homoscedasticity in the ordinary least squares estimation; therefore, overestimated variance can cause a bias toward the null hypothesis [17, 18]. Thus, confirmation of the spatial autocorrelation in neonatal care implies the need to address the spatial effects of unmeasured confounders in observational studies of neonates using nationwide real-world data.

This study aimed to investigate the geographical distribution of antimicrobial drug exposure among very preterm (VPT; gestational age 22–32 weeks) and very low birth weight (VLBW; <1,500 g) infants in Japan, with the objective of assessing its spatial effects.

## Methods

We conducted a retrospective study using the Diagnosis Procedure Combination (DPC) database, which is a Japanese administrative claims database. As of 2021, more than 2,000 hospitals have adopted the DPC-based reimbursement systems. This database contains anonymous clinical and administrative claims data, including baseline information, diagnoses, procedures, medications, and device use. Previous studies have confirmed the validity of this database [19]. Data were accessed on September 22, 2022. We evaluated the DPC database to identify admissions of infants, who were both VPT and VLBW, within the first two days of life from April 2010 to March 2021. We defined early neonatal drug exposure and neonatal drug exposure as any exposure to each class of drugs during the early neonatal period and neonatal period, i.e., within the first 7 days and 28 days of life, respectively. We defined two specific cohorts based on the discharge period of infants. Cases discharged during the early neonatal period were excluded. Consequently, we established an "early neonatal cohort" to specifically assess drug exposure during this phase. Similarly, upon the exclusion of cases discharged during the neonatal period, we delineated a separate "neonatal cohort" for further analysis. Infants discharged during the corresponding periods were excluded from the analysis. Prescriptions for these cohorts were extracted to calculate exposures to injectable antimicrobials classified as "antibacterial agents for systemic use" (J01), "antimycotic agents for systemic use" (J02), "antimycobacterial agents" (J04), and "antivirals for systemic use" (J05) according to the anatomical therapeutic chemical (ATC) classification [20, 21]. Nationwide early neonatal exposure and neonatal exposure to each class of drugs were determined. For international comparisons, exposures among extremely low birth weight infants were also assessed. We investigated the geographical distribution of antimicrobial drug exposure based on the prefecture where the

facility is located. Each of the 47 prefectures in Japan is an independent administrative entity that provides comprehensive neonatal care through cooperative perinatal care centers. Each prefecture has at least one medical school.

We examined the inter-prefectural spatial autocorrelation of early neonatal exposure and neonatal exposure to each antimicrobial agent using global and local Moran's *I* statistics [22]. We created local indicators of spatial association cluster plots and Moran scatter plots, which display each prefecture according to five categories: high-high, high-low, low-high, low-low, and not significant. A high-high classification indicates "high exposure surrounded by high exposures." This is indicative of a positive spatial autocorrelation or a spatial cluster of high exposures (a hot spot). A high-low classification is an outliner that indicates "high exposure in the target district while neighboring districts had low exposure." A low-high classification is another outliner that indicates "low exposure in the target district while neighboring districts had high exposure." A low-low classification indicates "low exposure surrounded by low exposures." Similar to the "high-high" scenario, this is indicative of positive spatial autocorrelation or a spatial cluster of low exposures (a cold spot). The "not significant" classification indicates a lack of either positive or negative significant spatial autocorrelation according to the local Moran's *I* statistics [15, 23]. Subsequent to the primary analysis based on the fifth level of the ATC classification (chemical substance), we conducted a sensitivity analysis using the higher fourth level of the ATC classification (chemical subgroup).

To explore the geographical heterogeneity underlying spatial autocorrelation, we aggregated the 47 prefectures into eight regions (Hokkaido, Tohoku, Kanto, Chubu, Kansai, Chugoku, Shikoku, and Kyushu regions) and generated heatmaps of the incidence of the early and late neonatal courses of antibacterial administration stratified by gestational age (22–23, 24–27, and 28–31 weeks), birth weight (<500, 500–999, and 1000–1499 g), and region. Late neonatal courses were defined as antibacterial prescriptions during the late neonatal period (between days 7 and 27 after birth [24]) after at least 7 antibacterial-free days. We also created heatmaps of the selection rates of each antibacterial agent on the first day of the antibacterial administration courses. Comparable heatmaps were created for early neonatal exposure to antifungal agents.

We computed weight matrices for spatial statistical analysis based on the *k*-nearest neighbors method, with *k* = 4 for centroid coordinates, because some prefectures are not contiguous. Spatial statistical analyses were performed using the *spdep* package in R, with the significance level set at 0.05 [25, 26]. Permission to conduct this study was obtained from the Ethics Committee of Tokyo Medical and Dental University (registration no. M2021-013). The study was exempted from informed consent due to the data being anonymized. We used open geographic data from the National Land Numerical Information released by the Japanese Ministry of Land Infrastructure, Transport and Tourism [27].

## Results

We identified 43,536 hospitalizations of VPT and VLBW infants within the first 2 days of life. We included 41,423 infants to analyze early neonatal drug exposure and 39,635 infants to analyze neonatal drug exposure (**Fig 1**). In the analysis of neonatal drug exposure, the median gestational age was 28.0 weeks (interquartile range: 26.0–30.0 weeks), and the median birth weight was 1,013 g (interquartile range: 760–1,253 g); male infants accounted for 52% of the cohort (**Table 1**). The mortality rates for all admissions, the early neonatal cohort, and the neonatal cohort were 6.8%, 3.8%, and 2.3%, respectively. The eight regions of Japan and the spatial weight matrix assigned to the 47 prefectures were visualized in **Fig 2**.

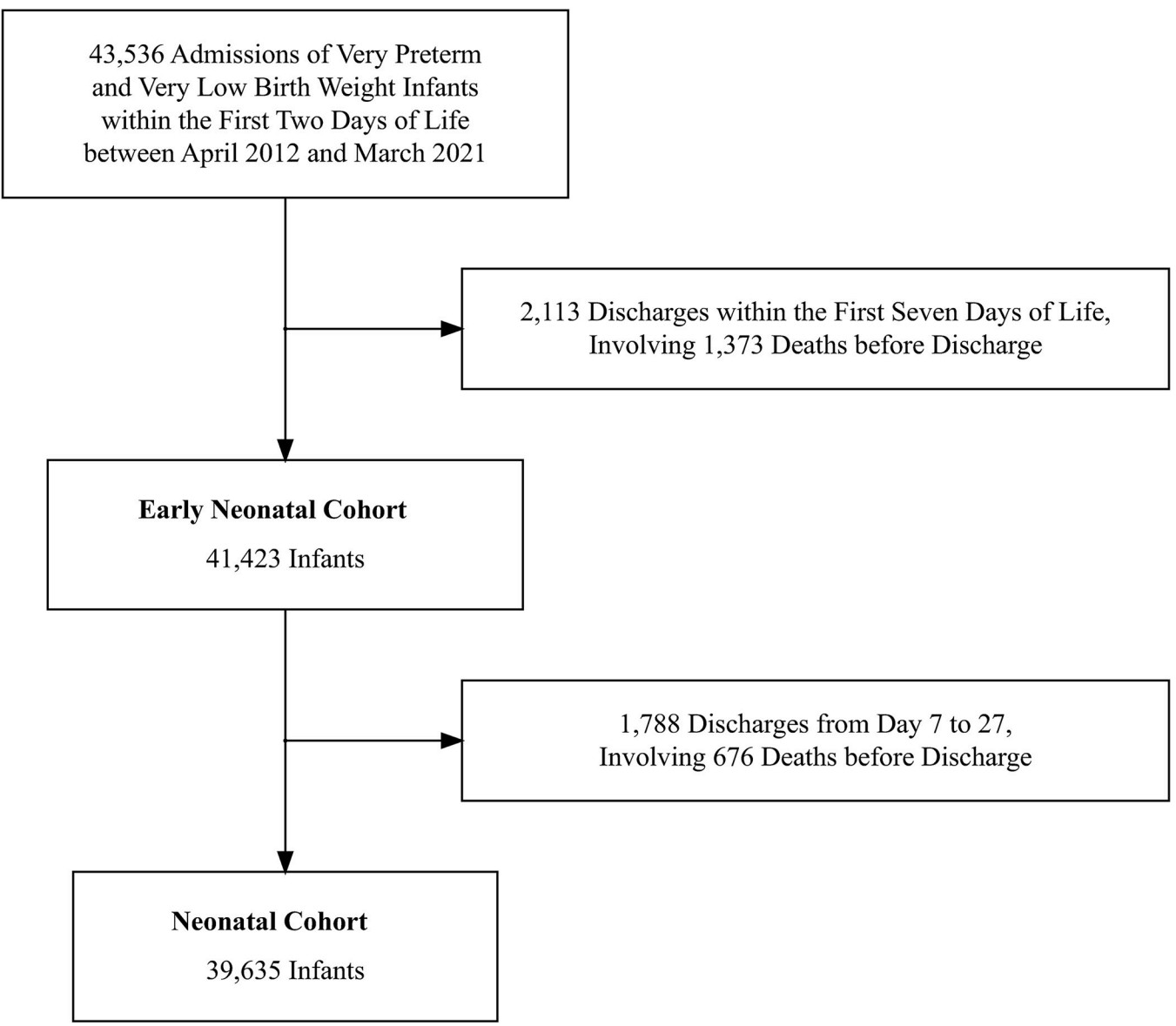

**Fig 1. Flowchart showing initial patient eligibility and exclusion because of early discharge.**

**Table 1. Patient characteristics.**

| Characteristic | All Admissions, N = 43,536[1] | Early Neonatal Cohort, N = 41,423[1] | Neonatal Cohort, N = 39,635[1] |
|---|---|---|---|
| Gestational Age (weeks) | 28 (26, 30) | 28 (26, 30) | 28 (26, 30) |
| Birth Weight (g) | 998 (740, 1,246) | 1,008 (754, 1,251) | 1,013 (760, 1,253) |
| Male | 22,692 (52%) | 21,556 (52%) | 20,636 (52%) |
| Inborn | 41,423 (95%) | 39,472 (95%) | 37,797 (95%) |
| Length of Stay (days) | 83 (61, 115) | 85 (64, 118) | 87 (66, 120) |
| Death before Discharge | 2,943 (6.8%) | 1,570 (3.8%) | 894 (2.3%) |

[1]Median (IQR); n (%)

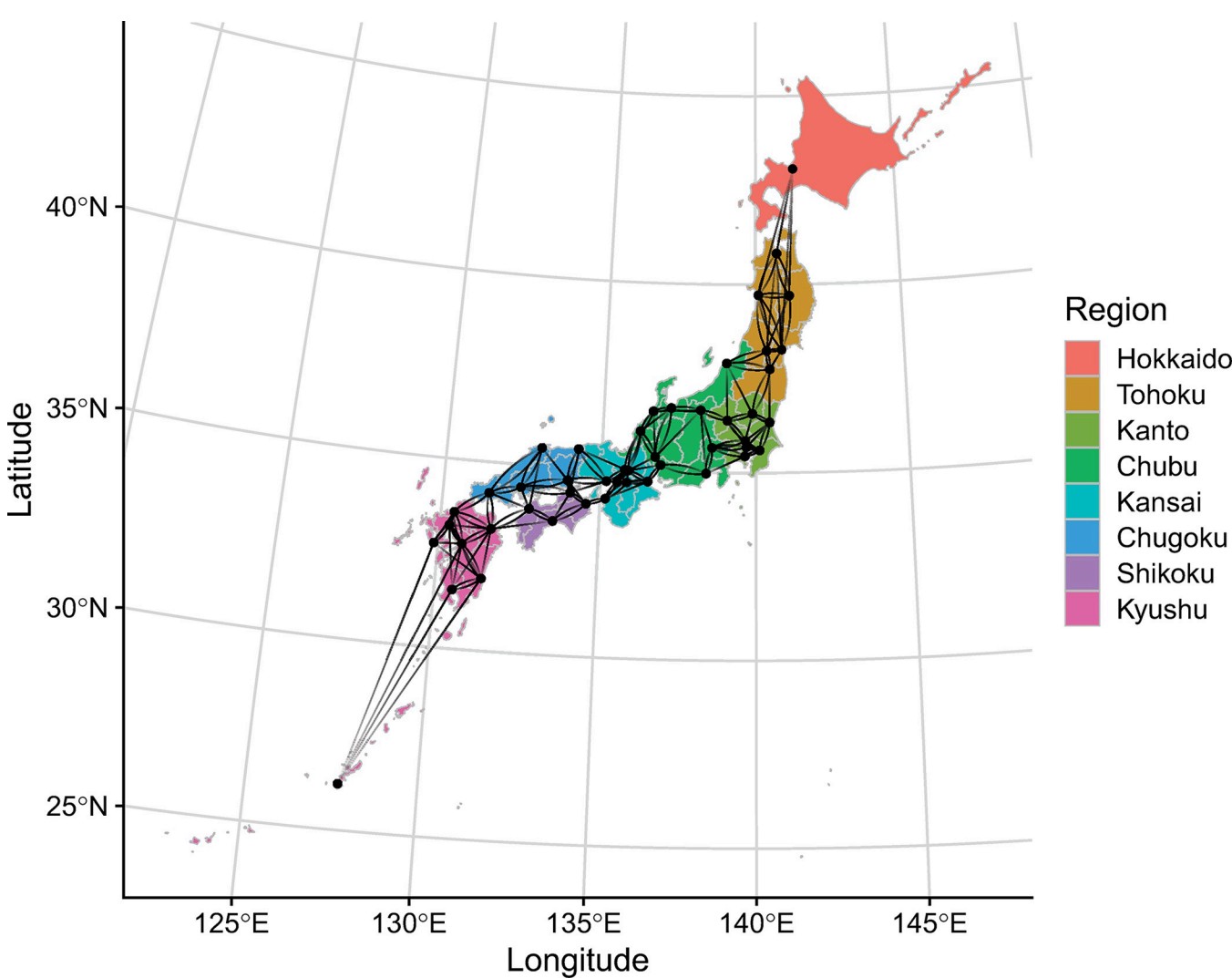

**Fig 2. Visualization of the eight regions of Japan and spatial weight matrix assigned to the 47 prefectures.**

Table 2 shows the frequently prescribed antimicrobials with a neonatal exposure rate of >10 per 1,000 infants. The top 10 antimicrobial agents prescribed during the neonatal period were as follows: ampicillin (J01CA01); amikacin (J01GB06); gentamicin (J01GB03); cefotaxime (J01DD01); fluconazole (J02AC01); ampicillin combinations (J01CA51); micafungin (J02AX05); cefmetazole (J01DC09); cefazolin (J01DB04); and vancomycin (J01XA01). These drugs had a neonatal exposure rate of >50 per 1,000 infants. Among these drugs, early neonatal exposure to fluconazole and neonatal exposure to amikacin, fluconazole, and cefmetazole showed significant positive spatial autocorrelations with the global Moran's *I* statistics. Local indicators of spatial association cluster plots revealed northern cold spots for early neonatal exposure and neonatal exposure to fluconazole in the Hokkaido and Tohoku regions (Fig 3). Higher early neonatal exposure and neonatal exposure to ampicillin, amikacin, gentamicin, and cefmetazole formed southern hot spots in the Kyushu region. Early neonatal exposure and neonatal exposure to each class of prescribed antimicrobials and their global Moran's *I* statistics are listed in S1 Table. The geographical distributions of antimicrobial drug exposure, local indicators of spatial association cluster plots, and Moran scatter plots for all prescribed drug

**Table 2. Frequently used drugs with early neonatal and neonatal exposure and their global Moran's *I* statistics.**

| Drug Class | Early Neonatal Cohort | | | Neonatal Cohort | | |
|---|---|---|---|---|---|---|
| | Exposure[1] | Moran's *I* | *P* Value[2] | Exposure[1] | Moran's *I* | *P* Value[2] |
| J01CA01. Ampicillin | 586.63 | 0.14 | 0.076 | 598.74 | 0.12 | 0.11 |
| J01GB06. Amikacin | 206.41 | 0.13 | 0.083 | 240.72 | 0.18 | 0.022* |
| J01GB03. Gentamicin | 190.18 | 0.11 | 0.13 | 196.54 | 0.10 | 0.2 |
| J01DD01. Cefotaxime | 137.87 | 0.06 | 0.4 | 162.86 | 0.05 | 0.4 |
| J02AC01. Fluconazole | 126.81 | 0.18 | 0.023* | 140.20 | 0.18 | 0.027* |
| J01CA51. Ampicillin, Combinations | 55.86 | 0.09 | 0.2 | 80.28 | 0.10 | 0.13 |
| J02AX05. Micafungin | 53.50 | -0.01 | >0.9 | 65.07 | -0.02 | >0.9 |
| J01DC09. Cefmetazole | 21.87 | -0.04 | 0.8 | 63.10 | 0.27 | <0.001*** |
| J01DB04. Cefazolin | 23.27 | -0.06 | 0.4 | 62.37 | -0.08 | 0.4 |
| J01XA01. Vancomycin | 10.24 | -0.03 | >0.9 | 57.70 | 0.08 | 0.2 |
| J01DH02. Meropenem | 11.83 | 0.12 | 0.11 | 34.41 | 0.09 | 0.2 |
| J02AB01. Miconazole | 25.40 | 0.20 | 0.002** | 27.48 | 0.20 | 0.002** |
| J01GB12. Arbekacin | 4.18 | 0.12 | 0.089 | 24.12 | 0.04 | 0.5 |
| J01DC14. Flomoxef | 15.02 | -0.04 | 0.7 | 23.89 | 0.03 | 0.5 |
| J01CA12. Piperacillin | 12.72 | 0.01 | 0.6 | 23.82 | 0.01 | 0.7 |
| J01FA01. Erythromycin | 15.26 | -0.09 | 0.3 | 23.04 | -0.09 | 0.3 |
| J01DD02. Ceftazidime | 9.56 | -0.02 | 0.9 | 22.48 | -0.03 | 0.8 |
| J01XA02. Teicoplanin | 3.60 | -0.04 | 0.7 | 16.63 | -0.02 | >0.9 |
| J01CR05. Piperacillin and Beta-Lactamase Inhibitor | 6.06 | -0.02 | 0.8 | 16.10 | -0.02 | >0.9 |
| J01DF01. Aztreonam | 15.72 | 0.18 | 0.001** | 16.02 | 0.18 | 0.001** |
| J01DH55. Panipenem and Betamipron | 8.23 | -0.04 | 0.8 | 13.55 | -0.05 | 0.7 |
| J01DE03. Cefozopran | 3.89 | 0.05 | 0.2 | 11.05 | 0.09 | 0.14 |
| J01GB01. Tobramycin | 7.05 | 0.15 | 0.002** | 8.63 | 0.22 | <0.001*** |
| J02AA01. Amphotericin B | 4.51 | -0.04 | 0.5 | 7.54 | -0.07 | 0.3 |
| J01XX08. Linezolid | 0.94 | -0.03 | 0.4 | 5.32 | -0.04 | 0.3 |
| J01XX01. Fosfomycin | 0.48 | 0.09 | 0.043* | 4.77 | -0.02 | >0.9 |
| J01DD62. Cefoperazone and Beta-Lactamase Inhibitor | 0.31 | -0.05 | 0.6 | 3.10 | -0.05 | 0.6 |
| J01DE02. Cefpirome | 1.06 | -0.04 | 0.4 | 2.75 | -0.04 | 0.4 |
| J01DC07. Cefotiam | 0.60 | 0.09 | 0.2 | 2.25 | -0.08 | 0.5 |
| J01FA10. Azithromycin | 1.57 | -0.03 | 0.8 | 2.09 | 0.01 | 0.6 |
| J01DH51. Imipenem and Cilastatin | 0.41 | 0.13 | 0.046* | 2.02 | 0.05 | 0.3 |
| J01FF01. Clindamycin | 0.29 | -0.07 | 0.5 | 1.72 | 0.05 | 0.3 |
| J02AX04. Caspofungin | 1.35 | -0.03 | 0.2 | 1.36 | -0.03 | 0.2 |
| J05AB01. Aciclovir | 0.87 | -0.01 | 0.8 | 1.31 | 0.02 | 0.5 |

[1]Exposure per 1,000 Infants

[2]Two-sided global Moran's *I* test

*$p<0.05$

**$p<0.01$

***$p<0.001$

classes are shown in **S1 Fig**. Prescriptions of antimycobacterial agents (J04) and antivirals (J05) were limited, and no remarkable spatial autocorrelation was detected. The sensitivity analysis based on a higher level of drug classification (**S2 Table** and **S2 Fig**) yielded comparable results. Our analysis of extremely low birth weight infants is displayed in **S3 Table**.

Heatmaps of regional diversity in drug administration stratified by gestational age and birth weight are presented in **S3 Fig**. No remarkable regional differences in the incidence of

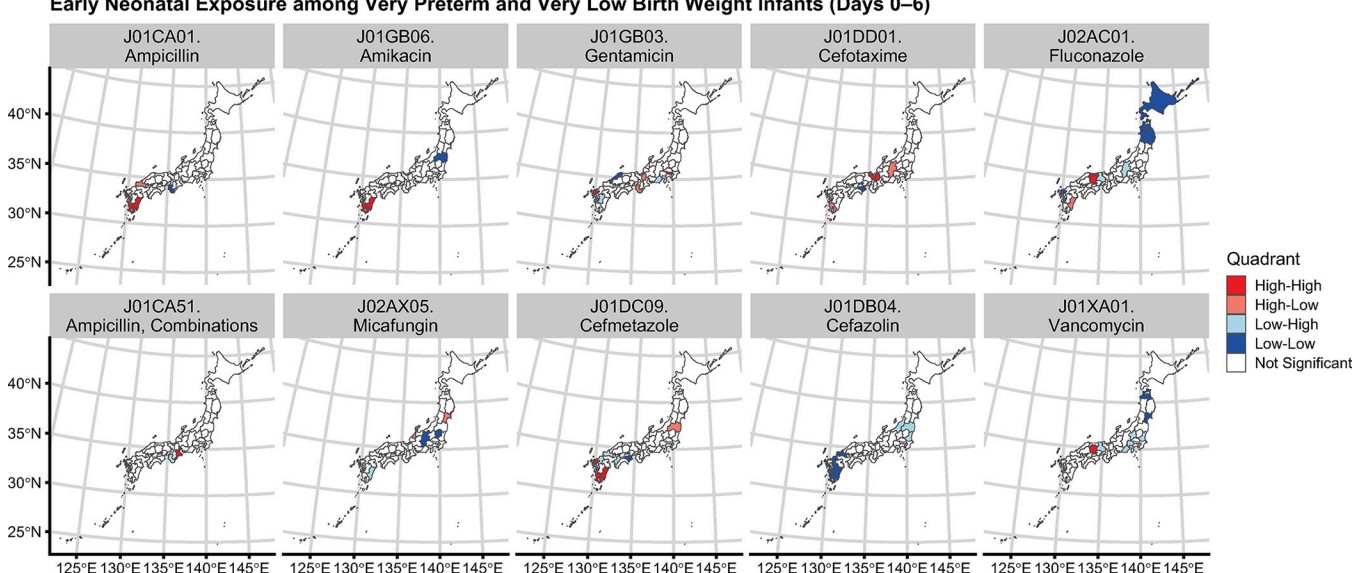

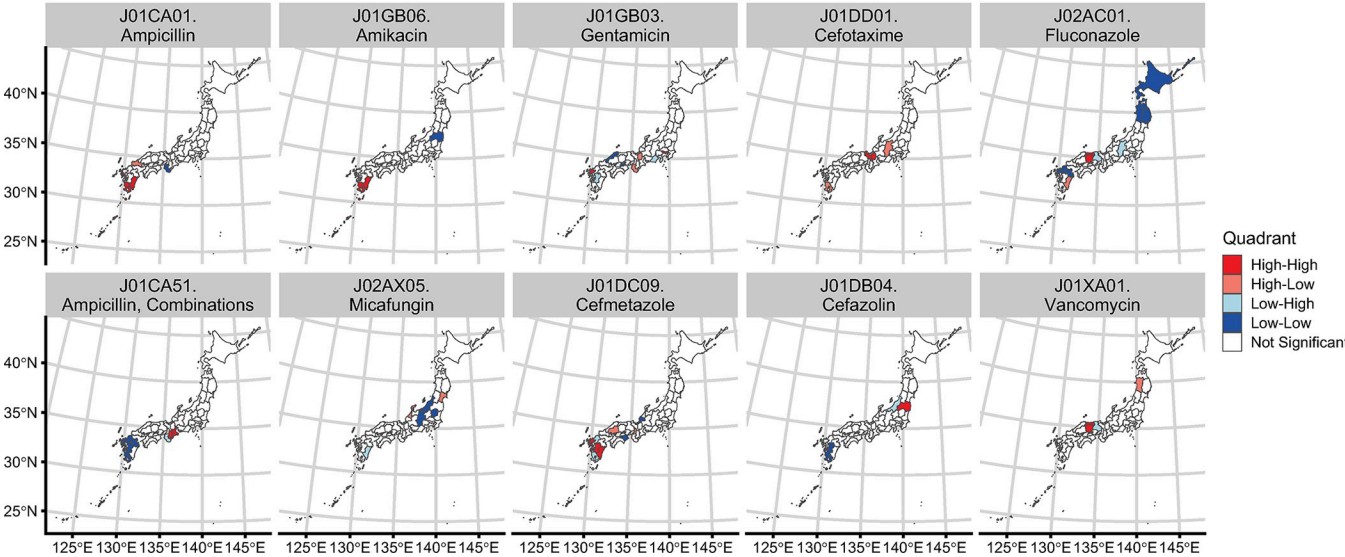

**Fig 3. Local indicators of spatial association cluster plots of early neonatal exposure and neonatal exposure to the top 10 frequently used antimicrobial agents.**

antibacterial drug administration were observed (**S3A Fig**). Regarding antibacterial drug selection during the early neonatal period, regional diversity was detected in the impact of gestational age on the administration of aminoglycosides (**S3B Fig**). Extremely preterm infants (gestational age <28 weeks) had significantly higher selection rates for aminoglycosides in the Hokkaido, Tohoku, Kanto, and Kansai regions (Fisher's exact test with Bonferroni correction). During the late neonatal period, higher selection rates for broad-spectrum antibacterial agents, such as fourth-generation cephalosporins, carbapenems, and glycopeptides, among extremely preterm infants were observed in some regions (**S3C Fig**). Early neonatal antifungal exposure was associated with extreme prematurity across all regions (**S3D Fig**). The Hokkaido and

Tohoku regions, which jointly formed a cold spot for fluconazole exposure, had the lowest antifungal use. Fluconazole exposure was also lower in the Kyushu region; however, overall antifungal exposure was not low, with miconazole and micafungin being prescribed instead.

## Discussion

This article is the first report on antimicrobial exposure and its geographical distribution among VPT and VLBW infants in Japan using a nationwide claims database. The most frequently used antimicrobials (listed in **Table 2**) were comparable to those used in the United States [7, 12]. The present study revealed positive spatial autocorrelations of early neonatal exposure and neonatal exposure to some antimicrobials (**Fig 3**). This research suggested no discernible regional differences in the incidence of antibacterial administration and indicated geographical heterogeneity in antibacterial and antimycotic drug selection (**S3 Fig**).

Empiric antibacterial therapy for neonates is performed at the discretion of the physicians after consideration of individual situations [12, 28]. In an international comparison focusing on extremely low birth weight infants, cefotaxime was the sole antimicrobial agent from the top 10 most frequently administered in the United States that exhibited higher exposure in Japan (190 vs. 126 per 1,000 infants). Cefotaxime, an alternative to ototoxic aminoglycosides for targeting gram-negative bacteria, saw higher selection rates in the Hokkaido region, where the selection rates of aminoglycosides were relatively low (**S3B Fig**). The preference for cefotaxime in Japan may reflect concerns about the adverse effects of aminoglycosides. The exposures to gentamicin and vancomycin in Japan were markedly lower than those in the United States (225 vs. 897 per 1,000 infants and 93 vs. 480 per 1,000 infants, respectively). Higher exposure to amikacin (301 vs. <43 per 1,000 infants), another class of aminoglycosides, may have reduced the exposure to gentamicin. In contrast, prescriptions of anti-methicillin-resistant *Staphylococcus aureus* agents other than vancomycin were limited, with arbekacin having the next-highest exposure level (37 per 1,000 infants), which may imply a lower risk of methicillin-resistant *Staphylococcus aureus* infection in Japan. These international variations indicate the necessity of incorporating local clinical practice patterns into drug research.

The global Moran's *I* statistics missed some of the spatial autocorrelations detected by the local Moran's *I* statistics. In the presence of geographical heterogeneity, spatial autocorrelation cannot be dismissed solely by global Moran's *I* statistics, which presumes equal variance [22]. Because the incidence of early-onset sepsis in VLBW infants is only 1% to 2%, the spatial effects of early neonatal antibacterial drug exposure can be attributed to the spatial effects of empirical antibacterial use [29]. Spatial effects of empirical treatment may reflect regional differences in the decision-making process of neonatologists facilitated by human interaction. A characteristic prescription pattern was identified in the Kyushu region, which has established a well-coordinated perinatal care system, including air transportation, to serve its island areas that comprise 4,600 islands [30]. Alternatively, spatial effects may result from environmental differences in antibiograms or other environmental factors. The unavailability of laboratory test results precluded the investigation of individual clinical backgrounds. Given the difficulties in measuring all confounders with spatial autocorrelation, a practical solution for nationwide observational research would be to adopt models that allow for residual spatial effects, such as Bayesian hierarchical modeling [31].

Decision-making involving antibacterial administration comprises two steps: evaluating whether the condition warrants an antibacterial prescription and selecting appropriate antibacterial agents. The lack of noticeable variations in the incidence of antibacterial administration suggests that the identified geographical heterogeneity in the selection of antibacterial agents was the primary cause of the spatial effects of antimicrobial exposure. Early neonatal

exposure to antifungal agents also showed regional differences. The nationwide trend toward administering antifungal agents mainly to extremely preterm infants is consistent with the prophylactic strategies recommended by previous studies [32].

## Limitations

The DPC claims database does not contain the actual dosage of drugs, medical indications, bacterial culture results, or other laboratory data. This information deficit has restricted exhaustive investigation of individual clinical backgrounds and regional infection trends, which may underlie regional differences in antimicrobial administration. Our study focused on identifying and analyzing drugs to which a notable proportion of infants were exposed at least once, and we did not evaluate the duration or total dosage of drug exposure. Similar to previous studies that have reported the prognosis of neonatal disseminated intravascular coagulation, approximately 5,000 VPT and VLBW infants were enrolled annually [33]. The national open data reported 62,154 births during this study period; therefore, it can be deduced that this study included approximately 70% of the infants in this category [34].

Reduced mortality observed after excluding infants who were discharged early (**Table 1**) suggested higher severity among the excluded infants. The rationale for this exclusion was the potential for missing prescriptions that would have been administered if the infants had survived and continued their hospitalization, which could lead to an underestimation of exposure. Selection bias caused by the exclusion of severe cases could be presumed to be limited because the comprehensiveness of our cohort was maintained. Of the 43,536 admitted infants, we included 41,423 (95.1%) in the early neonatal analysis and 39,635 (91.0%) infants in the neonatal analysis.

Transportations across prefectures may have resulted in a conservative assessment of positive spatial autocorrelation. Analyses using more granular geographic units would be greatly affected by spatial variations in patient backgrounds resulting from cross-boundary relocation facilitated by perinatal care collaboration systems. We must consider differences in perinatal healthcare systems when extrapolating our findings to other countries. The granularity of drug classification may also impact the results of statistical analyses. Mapping to the ATC system allowed reproducible multilevel drug classification. The consistent results across different ATC classification levels support the robustness of our study.

## Conclusion

Our study uncovered the commonly administered antimicrobial agents and the geographical distribution of antimicrobial drug exposure among VPT and VLBW infants in Japan, thereby revealing spatial autocorrelation and geographical heterogeneity. Further research is needed to determine the consequences of neonatal drug exposure. When conducting such investigations of neonatal care using nationwide real-world data, it would be desirable to employ models that can address its spatial effects.

## Supporting information

**S1 Checklist. STROBE statement—checklist of items that should be included in reports of observational studies.**
(DOCX)

**S1 Fig. Geographical distribution of early neonatal exposure and neonatal exposure to each antibiotic agent, tallied at the fifth level of the anatomical therapeutic chemical**

**classification.**
(PDF)

**S2 Fig. Geographical distribution of early neonatal exposure and neonatal exposure to each antibiotic agent, tallied at the fourth level of the anatomical therapeutic chemical classification.**
(PDF)

**S3 Fig. Heatmaps of regional diversity in drug administration stratified by gestational age and birth weight.** (A) Early and late neonatal episodes of antibacterial administration per 1,000 infants. (B) Drug selection rates on the first day of early neonatal courses of antibacterial administration (days 0–6). (C) Drug selection rates on the first day of late neonatal courses of antibacterial administration (days 7–27). (D) Early neonatal antimycotic drug exposure (days 0–6). We omitted the selection rate and exposure for categories with few cases, such as infants with a gestational age of 22 to 23 weeks and birth weight of 1,000 to 1,499 g and infants with a gestational age of 28 to 31 weeks and birth weight less than 500 g, because of concerns regarding high variance.
(PDF)

**S1 Table. Early neonatal and neonatal exposure and the global Moran's *I* statistics for each antimicrobial agent, tallied at the fifth level of the anatomical therapeutic chemical classification.**
(DOCX)

**S2 Table. Early neonatal and neonatal exposure and the global Moran's *I* statistics for each antimicrobial agent, tallied at the fourth level of the anatomical therapeutic chemical classification.**
(DOCX)

**S3 Table. Early neonatal and neonatal exposure among extremely low birth weight infants and the global Moran's *I* statistics for each antimicrobial agent.**
(DOCX)

## Acknowledgments

We would like to thank Editage for editing and reviewing this manuscript for the English language.

## Author Contributions

**Conceptualization:** Kota Yoneda, Daisuke Shinjo, Naoto Takahashi.

**Data curation:** Kota Yoneda.

**Formal analysis:** Kota Yoneda.

**Funding acquisition:** Daisuke Shinjo, Kiyohide Fushimi.

**Methodology:** Kota Yoneda, Daisuke Shinjo.

**Resources:** Kiyohide Fushimi.

**Software:** Kota Yoneda.

**Supervision:** Kiyohide Fushimi.

**Validation:** Kota Yoneda.

**Writing – original draft:** Kota Yoneda.

**Writing – review & editing:** Daisuke Shinjo, Naoto Takahashi, Kiyohide Fushimi.

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
