## [Decision Letter · Decision Letter 0]

5 Sep 2023

PONE-D-23-21704Geographical distribution of antimicrobial exposure among very preterm and very low birth weight infants: A nationwide database study in Japan.PLOS ONE

Dear Dr. Shinjo,

Thank you for submitting your manuscript to PLOS ONE. After careful consideration, we feel that it has merit but does not fully meet PLOS ONE’s publication criteria as it currently stands. Therefore, we invite you to submit a revised version of the manuscript that addresses the points raised during the review process.

We look forward to receiving your revised manuscript.

Kind regards,

Atalay Mulu Fentie, MPharm, PhD candidate

Academic Editor

PLOS ONE

3. We note that Figures 2, 3, S1 Fig and S2 Fig in your submission contain [map/satellite] images which may be copyrighted. All PLOS content is published under the Creative Commons Attribution License (CC BY 4.0), which means that the manuscript, images, and Supporting Information files will be freely available online, and any third party is permitted to access, download, copy, distribute, and use these materials in any way, even commercially, with proper attribution. For these reasons, we cannot publish previously copyrighted maps or satellite images created using proprietary data, such as Google software (Google Maps, Street View, and Earth). For more information, see our copyright guidelines: http://journals.plos.org/plosone/s/licenses-and-copyright.

A. You may seek permission from the original copyright holder of Figures 2, 3, S1 Fig and S2 Fig to publish the content specifically under the CC BY 4.0 license. 

B. If you are unable to obtain permission from the original copyright holder to publish these figures under the CC BY 4.0 license or if the copyright holder’s requirements are incompatible with the CC BY 4.0 license, please either i) remove the figure or ii) supply a replacement figure that complies with the CC BY 4.0 license. Please check copyright information on all replacement figures and update the figure caption with source information. If applicable, please specify in the figure caption text when a figure is similar but not identical to the original image and is therefore for illustrative purposes only.

Additional Editor Comments:

On the short title: correct the spelling error(wight) .. replace it with weight.

Please delete- "Compared to the results of previous studies in the United States, exposures to

cefotaxime and gentamicin were higher and lower in Japan, respectively." in the abstract section. You are not supposed to discuss results in the abstract section.

Please check all the references once again and shall be as per PLOS ONE guideline(e.g. put the "." after the intext citation).

Please cite the source of the late neonatal period definition. Also please, operationally define the early neonatal and neonatal cohort in the method section.

How do you define antimicrobial exposure (single dose vs multiple doses for prolonged time)? Was the study addressed duration? If not it shall be addressed as a study limitation.

As a limitation, clinical indications were not collected and analyzed. But on the conclusion, the geographical variation might be because of the clinical hetrogeniety. How do you come up with such conclusion. Can you please clarify?

Reviewers' comments:

Reviewer's Responses to Questions

**Comments to the Author**

1. Is the manuscript technically sound, and do the data support the conclusions?

Reviewer #1: Yes

Reviewer #2: Yes

2. Has the statistical analysis been performed appropriately and rigorously? 

Reviewer #1: Yes

Reviewer #2: Yes

3. Have the authors made all data underlying the findings in their manuscript fully available?

Reviewer #1: Yes

Reviewer #2: Yes

4. Is the manuscript presented in an intelligible fashion and written in standard English?

Reviewer #1: Yes

Reviewer #2: Yes

5. Review Comments to the Author

Reviewer #1: To the authors

The authors conduced a retrospective cohort study by using the Diagnosis Procedure Combination (DPC) database, to investigate the geographical distribution of antimicrobial drug exposure among very preterm (VPT) and very low birth weight (VLBW) infants in Japan, with the objective of assessing its spatial effects.

They showed the commonly administered antimicrobial agents and the geographical distribution of antimicrobial drug exposure among VPT and VLBW infants in Japan, thereby revealing spatial autocorrelation and clinical heterogeneity in the selection of antibacterial and antimycotic agents

This article might be an important paper, because that is the first report on antimicrobial exposure and its geographical distribution among VPT and VLBW infants in Japan using a nationwide claims database, and might lead to further research addressing how the spatial effects of neonatal care affect the outcomes of preterm infants in Japan.

Major problems

No major problems

Minor problems

All readers are not familiar with Global and local Moran’s I statistics. Although those might be basic things for statisticians, it would be better for the authors to have an additional explanation about what it means by high-high, high-low, low-low, and low-high in the selection of antimicrobials.

I am not sure that those statistical methods are appropriate for the explanation of the outcomes of preterm infants in Japan. Do the authors have any information of morbidity or mortality especially in Hokkaido or Kyushu area that might be different from other areas, because those regional differences in antimicrobial strategy might affect those outcomes?

Reviewer #2: Congratulations to the authors for this study that investigates the geographical distribution of antimicrobial exposure among 41,423 very preterm and very low birth weight infants admitted within the first two days of life in all 8 regions of Japan. It may be a model for other countries or regions to follow regarding the use of neonatal antibiotics in the first days after birth.

Abstract provides an informative summary of what was done and what was found.

Introduction describes appropriately the clinical problem with updated references related to the topic and justifies the study.

Results need modification of Figure 2 because is blurred, and Figure 3 is very blurred.

The discussion explains findings, limitations, and ideas for future research.

6. PLOS authors have the option to publish the peer review history of their article (what does this mean?). If published, this will include your full peer review and any attached files.

Reviewer #1: No

Reviewer #2: No

---

## [Author Response · Author response to Decision Letter 0]

8 Nov 2023

Our point-by-point response to all comments and suggestions is listed below:

We thank you for taking the time and effort necessary to review our manuscript and provide us with these valuable comments and suggestions. Accordingly, we revised our manuscript and made changes to it.

Journal requirements #1:

Response:

We thank you for the comment. We checked the style requirements.

---

Journal requirements #2:

In your Data Availability statement, you have not specified where the minimal data set underlying the results described in your manuscript can be found. PLOS defines a study's minimal data set as the underlying data used to reach the conclusions drawn in the manuscript and any additional data required to replicate the reported study findings in their entirety. All PLOS journals require that the minimal data set be made fully available. For more information about our data policy, please see http://journals.plos.org/plosone/s/data-availability.

Response:

We thank you for your comment. We have specified our contact information for accessing the data in the manuscript, referring to the "acceptable data access restrictions" section of the linked page and similar previous studies (Nagano, Hiroyuki, et. al., 2021. “Hospitalization for Ischemic Stroke Was Affected More in Independent Cases than in Dependent Cases during the COVID-19 Pandemic: An Interrupted Time Series Analysis.” PloS One 16 (12): e0261587., Ito, Fumiya, et. al., 2023. “Validation Study on Definition of Cause of Death in Japanese Claims Data.” PloS One 18 (3): e0283209.).

Page 1, lines 14–17 (Title page)

Data Availability: Data cannot be disclosed to the public due to a license agreement and ethical issues in each participating facility. For data sharing of the dataset generated during this study, please contact the corresponding author or the Office of Life Science and Bioethics Research Center (Email: infobec@tmd.ac.jp; Tel: +81-3-3813-6111). 

---

Journal requirements #3:

We note that Figures 2, 3, S1 Fig and S2 Fig in your submission contain [map/satellite] images which may be copyrighted. All PLOS content is published under the Creative Commons Attribution License (CC BY 4.0), which means that the manuscript, images, and Supporting Information files will be freely available online, and any third party is permitted to access, download, copy, distribute, and use these materials in any way, even commercially, with proper attribution. For these reasons, we cannot publish previously copyrighted maps or satellite images created using proprietary data, such as Google software (Google Maps, Street View, and Earth). For more information, see our copyright guidelines: http://journals.plos.org/plosone/s/licenses-and-copyright.

Response:

We have obtained written permission from the Ministry of Land, Infrastructure, Transport and Tourism to use content under the CC BY 4.0 license. Please check the uploaded document.

---

Journal requirements #4:

Response:

We thank you for the comment. Accordingly, we checked the reference list.

---

Editors’ Comments #1:

On the short title: correct the spelling error(wight) .. replace it with weight.

Response:

We thank you for pointing this out. Accordingly, we corrected the spelling.

Page 2, lines 24–25 (Title page)

Geographical distribution of antimicrobial exposure among very preterm and very low birth weight infants in Japan

---

Editor Comments #2:

Please delete- "Compared to the results of previous studies in the United States, exposures to cefotaxime and gentamicin were higher and lower in Japan, respectively." in the abstract section. You are not supposed to discuss results in the abstract section.

Response:

We thank you for this suggestion. Accordingly, we deleted the sentence.

---

Editors comment #3:

Please check all the references once again and shall be as per PLOS ONE guideline(e.g. put the "." after the intext citation).

Response:

We thank you for the suggestion. Accordingly, we checked the references and revised the in-text citations.

---

Editors’ comments #4:

Please cite the source of the late neonatal period definition. Also please, operationally define the early neonatal and neonatal cohort in the method section.

Response:

We thank you for the suggestion. We added the citation for the source of the definition (Oza S, Lawn JE, Hogan DR, Mathers C, Cousens SN. Neonatal cause-of-death estimates for the early and late neonatal periods for 194 countries: 2000-2013. Bull World Health Organ. 2015;93: 19–28). The original sentence “Infants discharged during the corresponding periods were excluded from the analysis." might have been unclear. Therefore, we elaborated on the description for clarity, as indicated below.

Page 5, lines 79–83, (Methods)

We defined two specific cohorts based on the discharge period of infants. Cases discharged during the early neonatal period were excluded. Consequently, we established an “early neonatal cohort” to specifically assess drug exposure during this phase. Similarly, upon the exclusion of cases discharged during the neonatal period, we delineated a separate “neonatal cohort” for further analysis.

---

Editors’ comments #5:

How do you define antimicrobial exposure (single dose vs multiple doses for prolonged time)? Was the study addressed duration? If not it shall be addressed as a study limitation.

Response:

We thank you for these queries. As noted in the method section, no distinction was made between single and multiple doses. The effect of cumulative drug exposure is a very important topic. However, susceptibility to adverse effects could vary widely based on genetic background, postnatal age, and postmenstrual age. In this study, we focused on identifying injectable antimicrobials prescribed for many infants. We added the text below to the limitation part.

Page 12–13, lines 230–232 (Limitation)

Our study focused on identifying and analyzing drugs to which a notable proportion of infants were exposed at least once, and we did not evaluate the duration or total dosage of drug exposure.

---

Editors’ comment #6:

As a limitation, clinical indications were not collected and analyzed. But on the conclusion, the geographical variation might be because of the clinical hetrogeniety. How do you come up with such conclusion. Can you please clarify?

Response:

We thank you for the suggestion. To avoid misunderstanding, the term “clinical heterogeneity” has been changed to “geographical heterogeneity” throughout the entire manuscript, including the following. The term “clinical heterogeneity” was intended to indicate heterogeneity in clinical care, not in the clinical backgrounds. Regional differences in the detailed treatment indication have not been verified. However, we have revealed the regional variations in clinical care of antimicrobial administration.

Page 13, lines 251–253 (Conclusion)

Our study uncovered …, thereby revealing spatial autocorrelation and geographical heterogeneity.

---

Reviewer #1’s comments #1:

All readers are not familiar with Global and local Moran’s I statistics. Although those might be basic things for statisticians, it would be better for the authors to have an additional explanation about what it means by high-high, high-low, low-low, and low-high in the selection of antimicrobials.

Response:

We thank you for the comment. We have revised the description as follows.

Page 5, lines 97–106 (Methods)

A high-high classification indicates “high exposure surrounded by high exposures.” This is indicative of a positive spatial autocorrelation or a spatial cluster of high exposures (a hot spot). A high-low classification is an outliner that indicates “high exposure in the target district while neighboring districts had low exposure.” A low-high classification is another outliner that indicates “low exposure in the target district while neighboring districts had high exposure.” A low-low classification indicates “low exposure surrounded by low exposures”. Similar to the “high-high” scenario, this is indicative of positive spatial autocorrelation or a spatial cluster of low exposures (a cold spot). The “not significant” classification indicates a lack of either positive or negative significant spatial autocorrelation according to the local Moran’s I statistics.

---

Reviewer #1’s comments #2:

I am not sure that those statistical methods are appropriate for the explanation of the outcomes of preterm infants in Japan. Do the authors have any information of morbidity or mortality especially in Hokkaido or Kyushu area that might be different from other areas, because those regional differences in antimicrobial strategy might affect those outcomes?

Response:

We thank you for the comment. As shown below, we have performed supplementary spatial analyses on early neonatal and infant mortalities in 2020 among very low birth weight infants in each prefecture, based on government statistics (https://www.e-stat.go.jp/en/node). No obvious spatial effects were identified. We are currently considering a causal inferential analysis based on linkage with a nationwide case registry. If required, we would add these figures to the supplementary materials. (You can find figures on "point_by_point_response_to_editor_and_reviewers.docx")

Fig. The early neonatal mortality and infant mortality among VLBW infants based on the Japanese government statistics.

---

Reviewer #2’s comments #1:

Results need modification of Figure 2 because is blurred, and Figure 3 is very blurred.

Response:

Thank you for the suggestion. The resolution of the raster image initially submitted was the maximum within the regulations. For better visibility, we created and uploaded these figures in EPS format.

---

Along with including the DOIs in the reference, the subsequent modifications were made:

Additional Corrections #1

We have corrected misstatements of the study period.

Page 3, line 33 (Abstract) and Page 5, lines 77 (Methods)

Before: from April 2012 to March 2021

After: from April 2010 to March 2021

---

Additional Corrections #2

In February 2023, the count of islands in Japan has changed due to enhancements in map precision, and we have reflected this accordingly.

https://www.gsi.go.jp/kihonjohochousa/pressrelease20230228.html

https://www.nippon.com/en/japan-data/h01615/

Page 12, line 213 (Discussion)

Before: (…) that comprise more than 2,500 islands [30].

After: (…) that comprise 4,600 islands [30].

---

## [Editor Report · Decision Letter 1]

22 Nov 2023

Geographical distribution of antimicrobial exposure among very preterm and very low birth weight infants: A nationwide database study in Japan.

PONE-D-23-21704R1

Dear Dr. Daisuke Shinjo

We’re pleased to inform you that your manuscript has been judged scientifically suitable for publication and will be formally accepted for publication once it meets all outstanding technical requirements.

Kind regards,

Atalay Mulu Fentie, MPharm, PhD candidate

Academic Editor

PLOS ONE
---

## [Editor Report · Acceptance letter]

29 Nov 2023

PONE-D-23-21704R1 

Geographical distribution of antimicrobial exposure among very preterm and very low birth weight infants: A nationwide database study in Japan. 

Dear Dr. Shinjo:

I'm pleased to inform you that your manuscript has been deemed suitable for publication in PLOS ONE. Congratulations! Your manuscript is now with our production department. 

Kind regards, 

on behalf of

Dr. Atalay Mulu Fentie 

Academic Editor

PLOS ONE